# Benefits of Cycling Wheelchair Training for Elderly with Physical Disability: A Prospective Cohort Study

**DOI:** 10.3390/ijerph192416773

**Published:** 2022-12-14

**Authors:** Jimmy Chun-Ming Fu, Pin-Kuei Fu, Yuan-Yang Cheng

**Affiliations:** 1Department of Physical Medicine and Rehabilitation, Taichung Veterans General Hospital, Taichung 40705, Taiwan; 2Department of Medical Research, Taichung Veterans General Hospital, Taichung 40705, Taiwan; 3College of Human Science and Social Innovation, Hungkuang University, Taichung 433304, Taiwan; 4Department of Post-Baccalaureate Medicine, College of Medicine, National Chung Hsing University, Taichung 40227, Taiwan; 5School of Medicine, National Yang Ming Chiao Tung University, Taipei 112304, Taiwan

**Keywords:** aged, disable, exercise, quality of life, wheelchair

## Abstract

Aim: In order to investigate the effect of cycling wheelchair training as an exercise for aged 65+ disabled patients on cognitive function, quality of life, aerobic capacity and physiological parameters. Methods: Participants in nursing home performed cycling wheelchair training for 30 min a day, 5 days a week, for a total of 4 weeks. The main outcome measure was the short form 12 survey (SF-12). Other outcome measures included the Mini-Mental State Examination (MMSE), aero bike work rate test, resting blood pressure, and heart rate. Results: In this study, 41 volunteers were recruited and no participants dropped out of the study voluntarily during training, and no serious adverse effect was identified. Physical and mental component summary total scores of SF-12 were significantly higher after training with statistical significance (*p* = 0.001). 8 subscales also showed significant improvements after training (*p* = 0.025 ~ <0.001). Total MMSE score has no difference before and after training. Attention/calculation (*p* = 0.018), short term memory (*p* = 0.041), and aerobic capacity (*p* < 0.001) as measured by subscales of MMSE and aero bike test showed marked improvements, while resting systolic blood pressure (*p* = 0.931) and heart rate (*p* = 0.793) did not change. Conclusions: Cycling wheelchair is practical for the disabled elderly to exercise, and a 4-week exercise program enhanced their quality of life and aerobic capacity.

## 1. Background

Aerobic exercise is beneficial for people of all ages [1,2,3,4,5]. Exercise can maintain body shape, reduce insulin resistance and improve blood lipid composition profile [6]. In the cardiovascular system, exercise has the effect of lowering blood pressure [7]. In the skeletal muscle system, exercise can prevent osteoporosis, lessen risks of fall and reduce the possibility of disability [8]. In addition, exercise can improve cognitive function, delay dementia, improve depressive mood and even lower risks of cancer [1,9,10,11]. As many countries in Far Eastern and Europe have entered or are about to enter an aging society, many adults aged 65+ lose the ability to walk due to factors such as disease, frailty, and aging. They need to use wheelchairs as a tool for mobility in most of their daily lives. Reduced mobility has negative impact on mental and physical health [12]. Environmental factors which act as facilitators and barriers also have crucial impact on accessibility and adherence of physical activities in wheelchair users [13,14]. Such wheelchair dependency makes them lose the possibility and accessibility to many kinds of exercise, thus keep them out of acquiring the benefits of exercise. However, the disabled elderly is the group that is particularly vulnerable to diseases. Enabling them to participate in exercise is therefore an important health issue.

At present, manual wheelchairs on the market require upper limbs of the user to wheel the chairs around. Although some exercise benefit can be obtained through these upper limb movements, muscle mass involved is much less than lower limb exercises, such as cycling, walking or jogging [15]. Hence, exercise through wheelchair propelling is less efficient than most exercises involving lower limbs [16]. On the other hand, the blood pressure and stress on the heart during upper limb exercise is relatively high. The heart rate increases much greater under the same exercise intensity with upper limbs when compared with lower limbs [17]. For subjects with a high risk of cardiovascular disease, precaution should be made before performing upper limb dominated exercise [18]. Furthermore, manual wheelchair users often experience musculoskeletal injuries related to shoulder overuse, and nerve injuries due to frequent compression around elbow and wrist joints. Examples are impingement injuries over the shoulder, cubital tunnel syndrome over the elbow, and carpal tunnel syndrome over the wrist [19,20]. Such injuries further lead to the loss of upper limb mobility and disturb self-care ability. In addition, for people with hemiplegia due to stroke or accidental injury, it is challenging for them to grasp the wheel rim and push the wheelchair with the upper limb on the affected side, resulting in restricted mobility.

The cycling wheelchair is an innovative design of wheelchair. The idea is to apply the mechanical principle of pedaling on a bicycle to a conventional wheelchair. The user activates large muscles of the lower limbs, such as quadriceps and gastrocnemius, to pedal and generate force to move the wheelchair. The accompanying steering system can be controlled at the same time by one hand to navigate the wheelchair. Compared with traditional wheelchairs, using lower limbs as a power source is more ergonomic and can avoid overuse injuries of the upper limbs [21]. The lower limbs have a higher overall muscle mass compared with upper limbs and are more suitable for aerobic exercise [15,16]. Theoretically, the user mobility of cycling wheelchairs would be greatly increased. In fact, a Japanese study reported that cycling wheelchair exercise induces more muscle activities of the paretic leg in patients with hemiplegic stroke and provides a chance of locomotion as well as a physical exercise [22].

The benefits of exercise for the elderly are well known. For elderly who must use a wheelchair as a tool for mobility, their exercise options are greatly reduced. Despite positive effects on physical fitness from upper extremity exercise had been reported [23], it is not easy for elderly people with poor muscle strength to propel a wheelchair solely by hands. It is even more difficult to rely on hand propelling a wheelchair to achieve doses of exercise required for fitness improvement [24]. Cycling wheelchairs is a potential exercise to help these elderly people with limited mobility to gain exercise benefits. In order to evaluate the therapeutic effect of cycling wheelchair exercise on the disabled elderly, we here designed this prospective cohort study to examine the effects on health-related quality of life, cognitive function, and aerobic capacity. We hypothesized that cycling wheelchair exercise improves the quality of life, cognitive function, and aerobic capacity for the disabled elderly.

## 2. Materials and Methods

### 2.1. Study Design and Participants

The study was a prospective cohort clinical study, which was approved by institutional review board of Taichung Veterans General Hospital (No. CE22294A) and retrospectively registered in ClinicalTrials.gov on 4 August 2022 (No. NCT05487898). Volunteers were recruited from a veterans’ nursing home from October 2021 to April. 2022. The research recruitment was announced with a poster on the bulletin board of the nursing home, and the inhabitants made the decision on whether to join the research by their own will. They were enrolled with the following inclusion criteria: (1) adults aged 65 or older, (2) mobility dependency on wheelchair, correspond to functional ambulation category scale (FAC) level 4 or below [25], and (3) ensure volunteer is capable of propelling wheel, muscle strength of both lower limbs is at least 4 rated by the Medical Research Council Scale (MRC) [26]. The exclusion criteria were: (1) having any acute medical problems within the past 6 months (2) having cardiopulmonary disease that could cause exercise hazards (3) unable to perform repetitive pedaling exercise due to diseases of the lower limbs (4) with pressure ulcers around hip and heel area (5) with impaired cognitive function that did not allow completion of our questionnaires.

### 2.2. Study Protocol

Each participant received a cycling wheelchair exercise training program, which was approved by an ethics committee of a medical center. The cycling wheelchair (DP-1608, COGY, Taipei, Taiwan) was designed based on the architecture of Kazunori Seki et al. [22]. During training, subjects pedaled the wheelchair with the rated perceived exertion 4–5 out of 10 in a flat courtyard of the veterans’ nursing home. A physical therapist accompanied the participant to ensure safety, and encouraged him to pedal the wheelchair to maintain the rated perceived exertion. The training session was conducted 30 min a day, five days a week, for a total of four weeks, as our previous research [27]. The actual training scene is presented as Figure 1.

### 2.3. Outcome Measures

Any adverse event during training program was recorded by training assistant. The goal of this study was to evaluate the therapeutic effect of cycling wheelchair. We recorded the following parameters one day before and one day after four weeks training. In order to evaluate health related quality of life, we used the short form 12 survey (SF-12). The SF-12 scale has 12 items under a total of 8 subscales. The subscales include general health perception (GH), physical functioning (PF), role limitation due to physical health (RP), bodily pain (BP), role limitations due to emotional problems (RE), vitality (VT), mental health (MH), and social functioning (SF). Scores of the 12 items were converted into the 8 subscales, which range from 0 (the worst) to 100 (the best). Two summary scores, the physical component summary (PCS) and the mental component summary (MCS), could then be calculated using the scores of the 8 subscales [28]. To standardize the PCS and MCS scores, we used a norm-based scoring algorithm that worked on data of the general population [29].

In order to measure cognitive function, we used the mini-mental state examination (MMSE), which evaluated mental abilities including orientation to time and place, registration, attention and calculation, short term memory, language skill, comprehension and follow instruction, and visuospatial abilities [30]. The full score of MMSE is 30. A score of 25 or higher is considered to be normal. A total score < 24 is abnormal. To test the aerobic capacity, an incremental cycle ergometer protocol modified from previous study was conducted with an aero bike (Comfort R7 Horizon, Johnson, Wisconsin, WI, USA) before and after the training program [31]. The participants were asked to pedal at a speed of 60 rpm, and the pedaling resistance was increased by 1 resistance level every minute. The highest resistance level at which the pedal speed maintained at 60 rpm was recorded. The setting of aero bike test was illustrated as Figure 1. Physiological parameters including body mass index (BMI), resting heart rate and blood pressure were documented by height and weight scale (BSM 370, InBody, Seoul, Republic of Korea) and sphygmomanometer (HBP-9030, Omron, Kyoto, Japan) before and after the exercise training. Reliability of SF-12 and MMSE in Chinese was validated in previous research [32,33]. Questionnaires and aero bike test before and after training were assessed by the same physical therapist. Physiological parameters were collected by duty nurse of the nursing home.

### 2.4. Statistical Analysis

We used the G*Power software version 3.1.9.7^a^ to calculate the estimated sample size. With the alpha error equals 0.05 and the power set at 90%, the effect size was 0.62 according to the preliminary data of 15 subjects. We found that a minimum of 30 subjects were required for our study. Statistical analyses were performed using SPSS^b^ for Windows (SPSS Ver.20, IBM, NY, USA).

Comparison of non-normally distributed results was carried out using Wilcoxon signed rank test to evaluate different parameters before and after cycling wheelchair training. Statistical significance was set at *p* < 0.05.

## 3. Results

### 3.1. Participants’ Basic Demographics

From October 2021 to April 2022, a total of 41 volunteers were enrolled in the study. One of them was hospitalized due to acute infectious disease during the training period. A total of 40 subjects completed the wheelchair cycling training. Their age ranged from 66 to 95 years old, with an average of 84.7 ± 8.2 years. 29 of them were males, and 11 of them were females. No volunteer withdrew or poor cooperated during exercise program. All participants completed all training sessions, despite three minor adverse events occurred during the entire training process without major trauma. One subject accidentally slipped off the wheelchair, one subject hit a wall due to improper steering, and one subject developed shoulder discomfort during exercise. 

### 3.2. SF-12

Table 1 shows the scores of the MCS, PCS, and the 8 subscales before and after cycling wheelchair training. Both PCS and MCS increased significantly after training. The 8 subscales also showed significant improvements after training.

### 3.3. MMSE

There was no significant improvement in the overall MMSE score (Table 2). However, after four weeks of training, subscales of attention/calculation and short-term memory significantly improved. While the number of participants scored 0 in attention/calculation remained to be four after training, the number of people who scored 5 points increased from 18 to 22. Other subscales including orientation, registration, comprehension and behavior revealed no statistically significant change. 

### 3.4. Aero Bike Test

Before training, 87.5% of the subjects attained 68 watts or less. After one month of training, this percentage dropped to 40%. The number of subjects who did end up at 78 watts or above increased significantly from 12.5% to 60% (Table 3). The mean peak work rate also increased markedly from 62.75 ± 10.63 to 72.93 ± 14.25 after training (Table 4). 

### 3.5. Physiological Parameters

All physiological parameters, including BMI, heartbeat, systolic blood pressure, and diastolic blood pressure, failed to reach statistical differences after training (Table 4).

## 4. Discussion

In this study we explored the effect of cycling wheelchair exercise training in the adults aged 65+ with physical disabilities at a long-term care facility. Of the 40 participants, all successfully completed the training, indicating the cycling wheelchair exercise is a feasible way of intervention with good compliance. Furthermore, significantly improved attention/calculation, short term memory, aerobic capacity and quality of life can be brought by the cycling wheelchair exercise program.

Physical activity is documented to have a positive relationship with the quality of life in the adults aged 65+ [34]. High levels of physical activity are correlated with a better perception for their quality of life [35]. A previous study showed that social and emotional benefits are the key motivators for people to do exercise, and exercise in turn brings a positive emotional outcome to form a virtuous cycle [36]. Exercise is a crucial factor in the perception of quality of life [37]. Among the various kinds of exercises, merely regular walking is capable of achieving a high quality of life in the adults aged 65+ [38]. Previous research showed that subacute and chronic LBP, improvements >3.77 in MCS and >3.29 in PCS, can be considered clinically relevant [39], while Minimal Clinically Important Difference in total knee arthroplasty population are 1.8 for the PCS and 1.5 for the MCS score [40]. In our study, one month of cycling wheelchair training could improve 6.86 in PCS score and 1.95 in MCS score, which revealed significantly improve the perceived health status of participants. More importantly, no volunteer withdrawn or poor cooperated during training program, which indicate that cycling wheelchair training is a feasible intervention strategy for improving quality of life of residents in long-term care facilities. 

Oxygen uptake is widely considered as an important parameter to evaluate aerobic capacity. The linear relationship between work rate and oxygen uptake is well known [41]. Furthermore, the maximal oxygen uptake can be estimated based on submaximal work rate [42], which is also verified in the adults aged 65+ population [43]. In this study, we utilized work rate as measured by an aero bike to investigate the aerobic capacity. Before cycling wheelchair training, only 12.5% of the subjects were able to attain 78 watts or more. After training, this percentage significantly increased to 60%, which indicated that most of the participants had significantly improved aerobic capacity after the cycling wheelchair training. Some participants may fail to reach maximal work rate due to insufficient muscle endurance, and thus measured peak work rate is considered as submaximal work rate. However, maximal work rate can be calculated through positive correlation with submaximal work rate [44], which indirectly indicated the improvement of maximal aerobic capacity.

Several studies have reported positive effects of exercise on cognitive function in the adults aged 65+ population. Submaximal treadmill walking was proven to enhance neural efficiency in those with mild cognitive impairments [45]. Submaximal exercise on stationary bicycles also significantly improved verbal memory and speed of information processing [46]. Other forms of exercise, such as Karate training and eastern Asian martial arts, could bring improvement in memory [47], attention, resilience and motor reaction [48]. The mechanism behind may be related to the enhanced brain neuroplasticity [49] and improved neural efficiency [45] brought by exercise, which may subsequently improve cognitive function. Furthermore, elevated levels of serum brain-derived neurotrophic factor were also reported after 4 weeks stationary bicycle training [27]. Since brain-derived neurotrophic factor was shown to promote the growth and differentiation of new neurons and synapses [50], it can also be part of the reason that exercise can promote cognitive function. Although the positive cognitive effect of various types of exercises is well known, few of them are applicable to wheelchair users. Subscales of the MMSE is documented as a valid test of episodic memory [33]. In this study, although the total MMSE score is no difference after training, we showed that cycling wheelchairs as an exercise significantly enhanced subscales of attention/calculation and short-term memory in the MMSE test. Interestingly, the magnitude of progress varied across participants. The number of people who scored 0 before and after training was the same, while the number of people who got full marks increased. This finding suggests that exercise training is an effective strategy for people who have partially impaired calculation ability. For those who have completely lost their computing ability, the effect was minimal.

Regular exercise training was evidenced to reduce resting heart rates, resting systolic and diastolic blood pressures [51]. Such a phenomenon was, however, not observed in our study. The possible reason may include the effects of parasympathetic response on heart rate and blood pressure drop with advanced ages [52]. In addition, the training intensity set in this study was based on the rated perceived exertion instead of heart rate measurement. Since some of our participants had mild cognitive impairment, it is possible that the intensity of exercise did not reach the amount recommended by the American College of Sports Medicine [53], which may obscure the alternations of heart rate and blood pressure after exercise training. 

There are limitations in this study. First, we did not have any restrictions on the participants’ daily physical activities during our exercise training programs. We were only able to inform them to maintain their regular lifestyle without any additional exercise training. Second, the training intensity was set by the rated perceived exertion instead of heart rate, which may result in under-training in some of the participants with mild cognitive impairment. Third, the duty nurse who measured parameters before and after training may be different, which result in some measurement bias. However, the nurse in nursing home is well-trained and experienced. Measurement variability is considered small and does not alter the statistical result. Lastly, no control group was designed in this study because participants in nursing home recruited in this study were all in their chronic stable condition. Anyone with acute medical problems in the recent six months before joining our program had been excluded. There should be no reason for the chronic functional impairment to be spontaneously improved in the outcome measurement of our study. Study design focus on comparison with other exercise strategy can be considered, however, as mentioned in introduction section, wheelchair user has limited choice of exercise and hinder the feasibility of this study design.

## 5. Conclusions

Our study disclosed the significant benefits in quality of life and aerobic capacity after 4 weeks of intensive cycling wheelchair training. According to the results of our study, exercise in the form of cycling wheelchair is recommended for adults aged 65+ patients with physical disability. As we only tracked the immediate effects after exercise intervention, further studies are needed to track more long-term effects.

## Figures and Tables

**Figure 1 ijerph-19-16773-f001:**
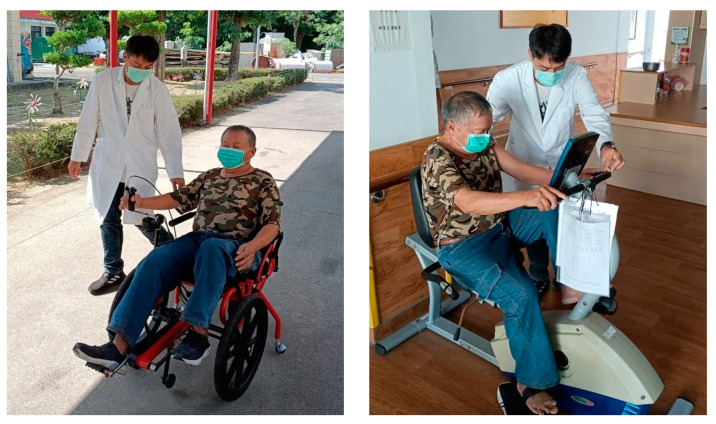
The scene of cycling wheelchair training (**left**) and the setting of aero bike test (**right**).

**Table 1 ijerph-19-16773-t001:** SF-12 survey before and after 4 weeks cycling wheelchair training.

	Before Training		After Training		*p* Value
Outcome Measure	Mean	SD	Mean	SD	
GH	40.00	22.50	51.88	19.10	<0.001 *
PF	41.25	31.80	48.75	30.98	0.003 *
RP	38.75	40.01	76.25	39.20	<0.001 *
BP	65.00	20.25	73.13	15.39	<0.001 *
RE	91.25	22.32	97.50	11.04	0.025 *
VT	57.50	18.22	65.50	16.32	<0.001 *
MH	76.50	11.22	83.00	8.53	<0.001 *
SF	58.75	20.84	67.50	20.57	<0.001 *
PCS-12	33.60	12.20	40.46	10.26	<0.001 *
MCS-12	54.60	4.54	56.55	2.72	0.001 *

GH, general health perception; PF, physical functioning; RP, role limitation due to physical health; BP, bodily pain; RE, role limitations due to emotional problems; VT, vitality; MH, mental health; SF, social functioning; PCS, physical component summary; MCS, mental component summary; SD, standard deviation; * *p* < 0.05.

**Table 2 ijerph-19-16773-t002:** MMSE before and after 4 weeks cycling wheelchair training.

	Before Training		After Training		*p* Value
Outcome Measure	Mean	SD	Mean	SD	
Orientation	6.25	2.01	6.40	1.95	0.465
Registration	2.90	0.38	2.78	0.53	0.16
Attention/Calculation	3.35	1.89	3.60	1.89	0.018 *
Short term memory	1.9	0.92	2.18	0.78	0.041 *
Comprehension/Behavior	5.05	1.26	4.93	1.36	0.132
Total score	23.95	5.56	24.10	5.65	0.527

SD, standard deviation; * *p* < 0.05.

**Table 3 ijerph-19-16773-t003:** The peak workload of aero bike test before and after cycling wheelchair training.

		Before Training		After Training	
Bike Resistance Levels	Work Load (W)	Number	Percentage	Number	Percentage
1	51	16	40.0	9	22.5
2	68	19	47.5	7	17.5
3	78	4	10.0	16	40.0
4	90	1	2.5	7	17.5
5	104	0	0	1	2.5
	Total	40	100.0	40	100.0

**Table 4 ijerph-19-16773-t004:** The physiological parameters before and after 4 weeks cycling wheelchair training.

	Before Training		After Training		*p* Value
Outcome Measure	Mean	SD	Mean	SD	
Work load(W)	62.75	10.63	72.93	14.25	<0.001 *
BMI (kg/m^2^)	23.48	3.42	23.53	3.49	0.104
Resting HR(beats)	76.59	8.04	75.62	8.98	0.793
SBP(mmHg)	126.97	11.54	126.90	11.87	0.931
DBP(mmHg)	69.17	9.35	71.03	12.35	0.411

BMI, body mass index; SD, standard deviation; * *p* < 0.05.

## Data Availability

Not applicable.

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
