# Peer review of "Benefits of Cycling Wheelchair Training for Elderly with Physical Disability: A Prospective Cohort Study"

_ijerph, 2022, doi:10.3390/ijerph192416773_

Round 1
Reviewer 1 Report
Dear Authors,
besides the intrinsic limitations of monocentric design, the topic is interesting considering the need for practicing aerobic physical exercise in people that use wheelchairs.
In this scenario, cycling wheelchair could be suitable tool to overcome barriers to physical exercise and improve mobility.
However, I have concerns about the implant of the study and some issues should be addressed to improve the paper.
Major Revisions
BACKGROUND: Page 1, line 31. “Aerobic exercise is beneficial for people of all ages”. As a general statement, some supporting evidence should be cited.
BACKGROUND: In my opinion, this section is well structured and well depicts the role of physical exercise as a preventive strategy in aging populations. However, in the complex topics of disability, the Background section should be implemented in consideration of architectural barriers, which set a well-known impairment in disabled patients’ mobility. Regarding this topic, you might consider some papers here enclosed.
- Lippi, L.; de Sire, A.; Folli, A.; Turco, A.; Moalli, S.; Ammendolia, A.; Maconi, A.; Invernizzi, M. Environmental Factors in the Rehabilitation Framework: Role of the One Health Approach to Improve the Complex Management of Disability. Int. J. Environ. Res. Public Health 2022, 19, 15186. https://doi.org/10.3390/ijerph192215186
- Ellison C, Struckmeyer L, Kazem-Zadeh M, Campbell N, Ahrentzen S, Classen S. A Social-Ecological Approach to Identify Facilitators and Barriers of Home Modifications. Int J Environ Res Public Health. 2021 Aug 18;18(16):8720. doi: 10.3390/ijerph18168720. PMID: 34444467.
- Wołoszyn N, Grzegorczyk J, Wiśniowska-Szurlej A, Kilian J, Kwolek A. Psychophysical Health Factors and Its Correlations in Elderly Wheelchair Users Who Live in Nursing Homes. Int J Environ Res Public Health. 2020 Mar 5;17(5):1706. doi: 10.3390/ijerph17051706. PMID: 32150994; PMCID: PMC7084309.
METHODS: Did the preliminary and post-training assessment take place according to a defined schedule (for instance, one day before and one day after). Please, provide a better definition of the timings.
METHODS: Were the questionnaires administered by the same operator before and after the training protocol? Moreover, could you please define which healthcare professional figure was involved in collecting the questionnaires? Was a defined MMSE cutoff set as exclusion criteria? Finally, were participants with heel pressure ulcers enrolled as well?
RESULTS: In my opinion, in order to increase the quality of the study, it could be useful to characterize the participants according to the pathology that determines their need to use the wheelchair. This could serve to further stratify the study results. Moreover, further description of participants’ medical history should be provided.
RESULTS: Were physiological parameters collected by the same operator?
DISCUSSION: The section should be implemented discussing the potential sources of bias encountered. For instance, whether pre-training vs post-training and physiological parameters assessments were performed by the same operator could influence your findings. Moreover, it should be noted that daily therapy, and changes in medication administration, could represent potential confoundment sources, as both cognitive and functional assessments could be consequently altered.
Minor Revisions
WHOLE MANUSCRIPT: Please, be sure to make each used acronym explicit along with its first mention in the manuscript (i.e. abstract, PCS and MCS, etc.).
WHOLE MANUSCRIPT: Please, correct spacing issues (i.e. page 1, line 23 “training(p”; page 1, line 33 “system[1]” etc.).
ABSTRACT: As requested by the journal’s authors guideline, the abstract should be organized into 4 sections: background, methods, results, and conclusion.
ABSTRACT: Page 1, line 13, line 15. Please, correct “aim” with “aims”, and “perform” with “performed”.
BACKGROUND: Page 1, lines 31-33 “[…] that involves”: it seems this period is structured with the wrong connector. Page 1, lines 33-34 “Exercise and lowering blood pressure”. Please, review the syntax.
BACKGROUND: Page 1, line 37. Please, correct “far eastern and europe” with “Far Eastern and Europe”.
METHODS: Page 2, line 90. Please, correct “during October” with “from October”.
METHODS: Page 3, line 113. Please, correct “is” with “was”.
DISCUSSION: Page 5, line 204. Please, correct “can improve” with “could improve”.
DISCUSSION: Page 6, lines 217-221; lines 236-239. “Although […] evaluation”, “In this study […] MMSE test”. Please, review these periods’ syntax.
DISCUSSION: Page 6, line 241. Please, correct “suggested” with “suggests”.
DISCUSSION: Page 6, line 259. Please, correct the typo (i.e. “,.”).
Author Response
As attachment file

Reviewer 2 Report
The paper is well written and well structured. It is easy to read. The problem is clearly stated. Participant recruitment, trial protocols and assessment tools are well presented.
However, I would have liked to have seen a brief example of each questionnaire in the appendix, as well as photos of the devices used in the study.
Also the tools and results are clearly presented, but I would have appreciated some additional figures to illustrate the results.
The conclusion is well supported and the limitations of the study clearly described.
Some comments:
- There are quite a few small typographical errors: missing or extra space.
On lines 128-129 the authors refer to a modified version of a protocol from a previous study. It would be interesting to know the modifications made.
Author Response
As attachment file

Round 2
Reviewer 1 Report
Dear Authors,
in my opinion, the manuscript is interesting, and the results are intriguing.
You have significantly improved the paper during the revision process.
Therefore, in my opinion, the paper is now suitable for publication in this Journal.
Best regards